# Protein Supplementation in a Prehabilitation Program in Patients Undergoing Surgery for Endometrial Cancer

**DOI:** 10.3390/ijerph20085502

**Published:** 2023-04-13

**Authors:** Josep M. Sole-Sedeno, Ester Miralpeix, Maria-Dolors Muns, Cristina Rodriguez-Cosmen, Berta Fabrego, Nadwa Kanjou, Francesc-Xavier Medina, Gemma Mancebo

**Affiliations:** 1Department of Obstetrics and Gynecology, Hospital del Mar, E-08003 Barcelona, Spaingmancebo@psmar.cat (G.M.); 2Campus del Mar, Universitat Pompeu Fabra, E-08003 Barcelona, Spain; 3Department of Endocrinology, Hospital del Mar, E-08003 Barcelona, Spain; 4Department of Anesthesia, Hospital del Mar, E-08003 Barcelona, Spain; 5FoodLab & UNESCO Chair on Food, Culture, and Development, Faculty of Health Sciences, Open University of Catalonia, E-08018 Barcelona, Spain

**Keywords:** prehabilitation, ERAS, endometrial cancer, surgery, protein supplementation

## Abstract

Enhanced recovery after surgery (ERAS) and prehabilitation programs are multidisciplinary care pathways to reduce stress response and improve perioperative outcomes, which also include nutritional interventions. The aim of this study is to assess the impact of protein supplementation with 20 mg per day before surgery in a prehabilitation program in postoperative serum albumin, prealbumin, and total proteins in endometrial cancer patients undergoing laparoscopic surgery. Methods: A prospective study including patients who underwent laparoscopy for endometrial cancer was conducted. Three groups were identified according to ERAS and prehabilitation implementation (preERAS, ERAS, and Prehab). The primary outcome was levels of serum albumin, prealbumin, and total protein 24–48 h after surgery. Results: A total of 185 patients were included: 57 in the preERAS group, 60 in the ERAS group, and 68 in the Prehab group. There were no basal differences in serum albumin, prealbumin, or total protein between the three groups. After surgery, regardless of the nutritional intervention, the decrease in the values was also similar. Moreover, values in the Prehab group just before surgery were lower than the initial ones, despite the protein supplementation. Conclusions: Supplementation with 20 mg of protein per day does not impact serum protein levels in a prehabilitation program. Supplementations with higher quantities should be studied.

## 1. Introduction

Endometrial cancer is today the most common gynecological cancer in the European Union and in the United States [1]. Risk factors include medical and public health conditions such as obesity, diabetes, and polycystic ovary syndrome [2]. Diagnosis is usually carried out in the early stages, given the appearance of metrorrhagia. The primary treatment is surgery, such as hysterectomy and bilateral adnexectomy. These interventions are generally carried out through a minimally invasive approach [3]. In any case, it should be noted here that even minimally invasive surgery entails aggression for the patient’s physiology [4].

In this context, the ERAS (Enhanced Recovery After Surgery) protocol emerges to reduce surgical aggression, and to obtain an earlier recovery of the patient, reducing complications. This protocol includes recommendations for perioperative and postoperative management that have been widely described in the existing literature [5,6].

The concept of prehabilitation was also created to improve the results. The objective of the prehabilitation is to prepare the patient for surgical treatment and to improve functional capacity and metabolic reserves, including medical, physical, nutritional, and psychological interventions [7,8]. Prehabilitation programs include recommendations to increase protein intake, as it has been proved that the increase in protein consumption reduces the number and severity of postoperative complications [9]. Nonetheless, we must also highlight here that there is no clear guidance or conclusive information about the quantity of protein to intake in prehabilitation programs.

The WHO (World Health Organization, Geneva, Switzerland) recommendations for protein intake in healthy adults are 0.66 g/kg per day (OMS recommendations in 2007) [10]. This UN organization shows different recommendations for special groups, such as children and pregnant women, but not for situations such as preparing for surgery or having cancer or other special health conditions. 

Following this premises and the present lack of existing information, the aim of our study was to analyze whether supplementation with 20 g of daily protein would be sufficient to maintain serum albumin, prealbumin, and total protein levels during the prehabilitation period.

## 2. Materials and Methods

### 2.1. Design and Subjects of This Study 

A prospective pilot observational study of patients undergoing laparoscopic surgery for endometrial cancer was conducted for more than two years, between January 2018 and March 2020, at the Department of Obstetrics and Gynecology of the Hospital del Mar in Barcelona (Spain).

Eligible patients for the study were, uniquely, women diagnosed with endometrial cancer and suitable for laparoscopic surgical treatment. As exclusion criteria, we included patients who declined surgery, an inability or incapacity to give an informed consent, the fact of having a non-resectable disease, having a degree of cognitive deterioration limiting or impeding their adherence to the program, or having surgery via laparotomy. The nutritional intervention was a feature of the prehabilitation program implemented in our hospital department in January 2018, and is extensively described in a previous paper [11]. Patients treated previously (before January 2018) were used as a control group and were classified according to the preoperative program followed: ERAS program (Enhanced Recovery After Surgery Program, without prehabilitation nor nutritional intervention), and PreERAS program (conventional preoperative program, which included only preoperative studies and anesthesiologist evaluation, and used before the establishment of the ERAS protocol).

The prehabilitation program was explained to the patients during the first oncologic gynecological visit and began on that day. The program involved preoperative and postoperative periods, and was maintained until 8 weeks after surgery. The preoperative part, the one analyzed in this paper, ranges between 2 and 6 weeks.

### 2.2. Nutritional Prehabilitation Intervention

Patients in the prehabilitation program were screened for malnutrition with the Malnutrition Universal Screening Tool (MUST) test, where a score of 2 or more indicates high risk, a score of 1 indicates intermediate risk, and 0 indicates low risk for malnutrition [12]. In addition to the MUST score, serum total protein, albumin, and prealbumin were assessed at baseline. All patients with a MUST score of 2 or more, or with albumin levels below 3 g/dl, were also previously treated by a nutritionist, trying to revert the eventual malnutrition status.

All patients in the prehabilitation group received a nutritional education program involving food selection and meal planning patterns, including an easy and feasible list of recipes for the homemade creation of protein supplements (mainly shakes and smoothies, always created with natural and non-processed ingredients such as fruits, vegetables, dried fruits, etc.) and adapted to diabetic patients, if necessary. These recipes included around 20 g of protein per day. All patients involved were previously instructed to take those oral protein supplements daily, always 30 min after exercise training, to enhance muscle hypertrophy. Those protein supplements prescribed did not alter the normal protein intake during meals. The food selection was made according to the WHO recommendations for healthy adults.

Patients in the PreERAS or ERAS program did not receive nutritional advice about increasing the protein intake, nor the recipes or strategies to achieve this. They were only assessed with the MUST test and serum albumin levels at baseline regarding nutritional aspects.

### 2.3. Variables and Outcomes

We used medical registries to retrospectively collect demographic and clinical baseline information. The nutritional status of women involved in this study was evaluated with the following indicators: total serum protein, prealbumin levels, and albumin in the prehabilitation patients’ group. Measurements were recorded baseline at the time of the first visit, immediately before the surgery, and 24–72 h post surgery. Patients in the preERAS or the ERAS group were only evaluated at baseline and 24–72 h after surgery with serum protein and albumin, adding the prealbumin level in the ERAS group.

### 2.4. Statistical Methods

The statistical analysis applied in this study was performed using SPSS 25.0 (Chicago, IL, USA), accepting a statistically significant level of 5% (*p* < 0.05). Both the demographic and clinical characteristics of our patients were summarized using descriptive statistics. Continuous variables were reported as mean (range) or mean ± standard deviation (SD) when indicated. Categorical variables were reported as frequency and percentage (%). We used, when appropriate, the Pearson’s chi-square test or the Fisher’s exact test to compare efficiently categorical variables. We used the Student’s *t*-test or the non-parametric Mann–Whitney test to compare continuous variables. 

## 3. Results

A total of sixty-eight consecutive patients undergoing laparoscopy surgery for endometrial cancer were included in the prospective prehabilitation cohort of the study. In the historical cohorts, 60 patients were included in the ERAS group and 57 in the preERAS group.

The mean patient age in the prehabilitation group was 66.4 years (range, 35–86 years), with no differences with the two other groups. The median time of patients who followed the prehabilitation program before surgery, which includes the protein supplementation, was 25 days (Interquartile 18–35) days. The baseline demographic and clinical characteristics of the study population according to the perioperative program are shown in Table 1. The groups were comparable in baseline characteristics in terms of age, BMI, comorbidities, ASA, and cancer stage.

According to the MUST score, only six patients (9.8%) were at high risk of malnutrition (five patients scored 2, one scored 1). All other patients had a MUST score of 0 (90.2%).

The values of serum total protein, albumin, and prealbumin at baseline were similar in the three groups (Table 2), although there was a trend toward better values in the prehabilitation group. It should be noted that these values are previous to any intervention.

Results comparing the impact of the nutritional intervention program on blood analysis just before surgery and 24–72 h after we recorded it are in Table 2 and Figure 1. Values after surgery decreased in all three groups. The prehabilitation group was the only one which had blood tests before surgery. The values showed a decrease in serum albumin, prealbumin, and total proteins, even with the nutritional intervention. Differences in basal and after surgery data were not statistically significant between the three groups.

Table 2 also shows the variation between the values after surgery and baseline. Although the decrease was lower for all parameters in the prehabilitation group compared to the preERAS and ERAS groups, the difference was not statistically significant. 

## 4. Discussion

This research reported our experience with implementing protein supplementation intervention in a prehabilitation program for endometrial cancer patients undergoing laparoscopic surgery. We would like to assess the impact of this nutritional intervention on serum total protein, albumin, and prealbumin. The main results failed, nevertheless, to confirm a real nutritional improvement among women included in the prehabilitation group. 

Our prehabilitation programs consist of the earlier preparation of patients between diagnosis and surgery. This preparation allows the patients to improve functional capacity and metabolic reserves before surgical intervention. It includes physical, emotional, and nutritional interventions. Previous published papers confirmed that multimodal prehabilitation programs used in cases of major cancer surgeries show a positive impact on the patients’ outcomes [11,13]. Nutritional support comprises pre-operative carbohydrate loading just before surgery, and nutritional interventions aim to increase the protein intake [5].

The main objective of improving the nutritional status of the patients is its potential role in reducing perioperative and postoperative complications. In our study, serum levels of albumin and prealbumin were proposed for preoperative risk stratification [14]. In their review article, Loftus et al. show that low albumin and prealbumin levels are associated with an increase in surgical complications. In this regard, the authors address the inability to differentiate between malnutrition and acute inflammation, since those parameters are also altered because of inflammation. Nevertheless, and even having in mind those significant limitations, there are data that protein supplementation could decrease complication rates, as shown in the classical study from the Veterans Affairs’ where it was demonstrated that severely malnourished patients supplemented with parenteral nutrition in the perioperative period had better surgery outcomes (veterans). There are conflicting data about this topic. Van Venrooij [13] studied the complication rate in patients well-nourished undergoing cardiac surgery, showing no improvement in preoperative protein and energy intake.

Salvetti et al. showed that in patients undergoing elective spine surgery, a threshold of <20 mg/dl for prealbumin was correlated with an increase in surgical site infection (17.8% versus 4.8%) [15]. A meta-analysis and a systematic review performed by Liu showed that decreased preoperative albumin in patients undergoing surgery for urothelial carcinoma predicted poor overall survival, cancer-specific survival, recurrence-free survival, 30-day complications after surgery, and 90-day mortality after surgery [16]. Kabata also published a randomized control trial of preoperative nutritional support in cancer patients with no clinical signs of malnutrition and undergoing abdominal cancer surgery. Patients receiving nutritional supplementation for 14 days before surgery had a lower rate of postoperative complications. Serum total protein and albumin were stable compared to the control group, where the levels decreased [17].

Previously, we analyzed the role of nutritional prehabilitation in women with ovarian cancer [18]. We selected patients undergoing a prehabilitation program during neoadjuvant chemotherapy and interval cytoreductive surgery. The data showed an increased postoperative recovery and decreased intraoperative complications (40% vs. 14.3%) in the prehabilitation group.

Hamaker et al. published a systematic review of nutritional status in patients with cancer, showing values as high as 49% of patients malnourished [19]. There are no data about this in endometrial cancer specifically, but our data show low levels of malnourished patients. This is the reason why we supplemented diets with only 20 mg of protein a day and kept patients on their normal daily eating habits (which is the easiest way to obtain useful comparative results). Nevertheless, we must remark that our study failed to improve nutritional status, even with the mentioned protein supplementation. This was not, however, the only unexpected result; in addition, the serum nutritional status even decreased during the prehabilitation program before surgery, which was a very unexpected finding in the framework of this study.

Other studies have shown interesting results improving nutritional status with higher levels of supplementation. In this regard, ERAS protocols recommend protein intakes of 1.2–2.0 g/kg/day from high-quality protein sources [20,21], including normal diet and supplementation. In a randomized trial carried out by Kabata, the prescribed dose was 40 mg of protein per day. In this study, modest increases in albumin and total protein were found, while, however, in the control group, the levels decreased [17]. Following these premises, our recommendation was to half this intake quantity. Additionally, this is probably the main reason we have found no improvement in the intervention group.

One of the most unexpected results was that even with the protein supplementation, our patients had a decrease in the serum levels of total protein, albumin, and prealbumin. This variation occurred in only 25 days, in patients non-malnourished and with neoplasia with little systemic affectation. ESPEN guidelines (European guideline on obesity care in patients with gastrointestinal and liver diseases) recommend perioperative support only in patients with malnutrition or at nutritional risk, and do not provide any information about how they should be supplemented [9]. In our case, we found that even patients not at risk for malnutrition could also benefit from protein supplementation.

We have also to remark that one of the reasons why we have not observed any benefit in our study could be the period time during which the patients take the protein supplements. The mean time of protein supplementation in our series was 25 days, which could explain why no benefit was observed. However, in Kabata et al.’s study [17], with only 14 days of supplementation, they observed beneficial effects in terms of morbidity and stability in nutritional parameters. The protein supplementation was also 20 g of protein per day, although from a commercial preparation. We might think that the homemade natural shakes or smoothies are not as effective, perhaps due to bioavailability issues, as the commercial ones.

### Strengths and Weaknesses

This research also presents different strengths that we would like to remark on. As far as we know (and as far as we have been able to verify from the published literature), this is the first piece of research studying the evolution of nutritional parameters in patients with endometrial cancer during the prehabilitation and after-surgery periods. Additional strengths include this being a homogeneous study (both in demographics and clinical characteristics) regarding a population with endometrial cancer undergoing laparoscopic surgery.

Potential weaknesses of our research also include, on the other hand, the non-randomized control trial design, and the limitations usually associated with comparing results with retrospective cohorts. Another weakness is that we have not recorded the adherence of the patients to the recommended supplements, and the results could be a result of low adherence to them. Finally, we must note also that we could not address aspects that should affect results such as the bioavailability of the nutrients.

## 5. Conclusions

As we have highlighted throughout this article, the aim of this study was to assess the impact of protein supplementation (with 20 mg per day) before surgery in a prehabilitation program. This impact was supposed to affect postoperative serum albumin, prealbumin, and total proteins in endometrial cancer patients undergoing laparoscopic surgery.

A prospective study including patients who underwent laparoscopy for endometrial cancer was conducted, with a total of 185 patients divided into three groups (68 in the prehabilitation group). Surprisingly, there were no basal differences in serum albumin, prealbumin, or total proteins between the three groups. After surgery, and regardless of the type of nutritional intervention, the decrease in values was also similar. The values in the Prehab group before surgery were even lower than the initial ones, despite protein supplementation.

In view of all the above, we can conclude that supplementation with 20 mg of protein a day, in patients undergoing laparoscopic surgery for endometrial cancer, is not sufficient to maintain the levels of serum total protein, albumin, and prealbumin. Higher supplementation is necessary, considering that, even with this extra protein intake, the observed levels decreased.

## Figures and Tables

**Figure 1 ijerph-20-05502-f001:**
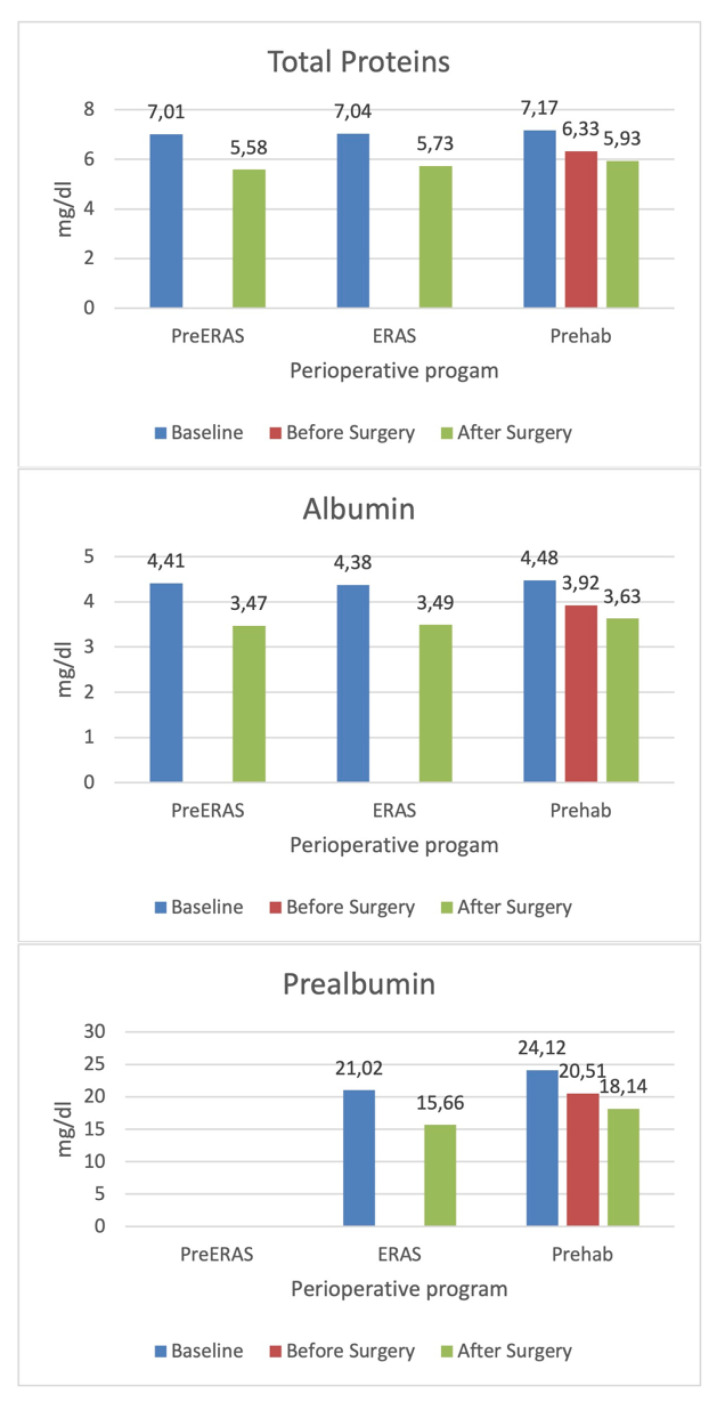
Variation of serum levels according to intervention groups.

**Table 1 ijerph-20-05502-t001:** Baseline patients’ characteristics according to the followed perioperative program.

	PreERAS (n = 57)	ERAS (n = 60)	Prehab (n = 68)	*p* Value
Age (years), mean [range]	64.1 [39–88]	67.4 [44–92]	66.4 [35–86]	0.310
BMI kg/m^2^, mean (SD)	32.0 ± 7.1	29.1 ± 6.5	31.0 ± 7.1	0.062
Smoking, n (%)	14 (24.6)	9 (15.0)	9 (13.2)	0.211
Hypertension, n (%)	31 (54.4)	37 (61.7)	40 (58.8)	0.724
Dyslipidemia, n (%)	18 (31.6)	21 (35.0)	23 (33.8)	0.924
Diabetes, n (%)	13 (22.8)	11 (18.3)	15 (22.1)	0.813
ASA, n (%)				0.578
I	7 (12.3)	4 (6.7)	5 (7.4)	
II	36 (63.2)	39 (65.0)	51 (75.0)	
III	14 (24.6)	16 (26.7)	11 (16.2)	
IV	0	1 (1.7)	1 (1.5)	
Disease Stage (FIGO), n (%)				0.496
IA-IB	41 (71.9)	43 (71.7)	56 (82.4)	
II	7 (12.3)	7 (11.7)	7 (10.3)	
IIIA-IIIC	9 (15.8)	10 (16.7)	5 (7.4)	
IV	0	0	0	

ASA: American Society of Anesthesiologists, BMI: body mass index, SD: standard deviation.

**Table 2 ijerph-20-05502-t002:** Nutritional parameter value evolution according to the followed perioperative program (preERAS vs. ERAS vs. Prehab).

		PreERAS (n = 57)	ERAS (n = 60)	Prehab (n = 68)	*p* Value
Baseline	Total proteins (g/dL)	7.01 ± 0.42	7.04 ± 0.40	7.17 ± 0.45	0.076
	Albumin (g/dL)	4.41 ± 0.29	4.38 ± 0.25	4.48 ± 0.33	0.138
	Prealbumin (mg/dL)	NA	21.02 ± 4.16	24.12 ± 6.95	0.288
Preoperative	Total proteins (g/dL)	NA	NA	6.33 ± 0.70	NA
	Albumin (g/dL)	NA	NA	3.92 ± 0.46	NA
	Prealbumin (mg/dL)	NA	NA	20.51 ± 5.29	NA
Postoperative	Total proteins (g/dL)	5.58 ± 0.57	5.73 ± 0.57	5.93 ± 0.66	0.085
	Albumin (g/dL)	3.47 ± 0.31	3.49 ± 0.36	3.63 ± 0.43	0.128
	Prealbumin (mg/dL)	NA	15.66 ± 4.17	18.14 ± 4.74	0.194
Decrease	Total proteins (g/dL)	1.49	1.30	1.23	0.376
	Albumin (g/dL)	0.93	0.89	0.85	0.758
	Prealbumin (mg/dL)	NA	5.73	5.82	0.970

NA: non applicable.

## Data Availability

The data presented in this study are available on request from the corresponding author. The data are not publicly available due to privacy reasons.

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
