# Peer review of "Protein Supplementation in a Prehabilitation Program in Patients Undergoing Surgery for Endometrial Cancer"

_ijerph, 2023, doi:10.3390/ijerph20085502_

Round 1

Reviewer 1 Report

The paper entitled Protein supplementation in a Prehabilitation Program in patients undergoing surgery for endometrial cancer” presented by Sole-Sedeno et cols intended to assess the impact of protein supplementation with 20mg per day before surgery in a prehabilitation program in postoperative serum albumin, prealbumin, and total proteins in endometrial cancer patients undergoing laparoscopic surgery.   

These are our suggestions for improvements in the paper:

1 – On line 62 (page 1) there is something between parentheses (REF) that is not connected with the text. We believe should be include a number for a reference there.

2 – On line 81 (page 2) it is not clear to what “this information” is related to. There are too much information in the previous paragraph, so the reader get confused about to which information you meant to. 

3 – It was not described the time of prehabilitation protocol (probably in the previous paper published that is not listed). The time of exposition could explain the lack of results

4 – The anthropometric profile shows that participants had different degrees of overweight. Considering that absorption of protein is higher when it is needed maybe the supplementation in well-nourished participants could not be so effective as in the unnourished ones ant this  could help to explain the results.

5 – It is not described if the supplementation was made by regular diet or with some nutritional supplements. It is clear that biodisponibility of nutrients is different when supplemented isolated or in combination with other nutrients.

6  - We believe that conclusions about the supplementation dose cannot be inferred because variables like biodisponibility and time of suplementation

Author Response

Dear Reviewer,

Thank you very much for your valuable work and also for the comments to the article, that will help to improve the text.

Please, find here the answer to all your comments, one by one:

"1 – On line 62 (page 1) there is something between parentheses (REF) that is not connected with the text. We believe should be include a number for a reference there."

Solved, sorry for the mistake.

“2 – On line 81 (page 2) it is not clear to what “this information” is related to. There are too much information in the previous paragraph, so the reader get confused about to which information you meant to.”

 The paragraph has been rewritten to increase clarity in that way:

Patients in the PreERAS or ERAS program didn’t receive nutritional advice about increasing the protein intake, nor the recipes or strategies to achieve that. They were only assessed with the MUST test and serum albumin levels at baseline regarding nutritional aspects.

"3 – It was not described the time of prehabilitation protocol (probably in the previous paper published that is not listed). The time of exposition could explain the lack of results "

We add a paragraph in the methods sections explaining that:

The prehabilitation program was explained in the first gynecological visit and began on that day. The program involved preoperative and postoperative periods and was maintained until 8 weeks after surgery. The preoperative part, the one analyzed in this paper, ranges between 2-6 weeks.

We add a sentence in the results part, line 127, to clarify that:

The median time of patients who followed the prehabilitation program before surgery, which includes the protein supplementation,  was 25 days (Interquartile 18-35) days

“4 – The anthropometric profile shows that participants had different degrees of overweight. Considering that absorption of protein is higher when it is needed maybe the supplementation in well-nourished participants could not be so effective as in the unnourished ones ant this  could help to explain the results.”

Endometrial cancer patients usually are patients well-nourished. It is a cancer diagnosed in early stages, and the only symptom in light vaginal bleeding. As you well said, they are more overweight than well-nourished. There are extensive data about that and it is one of the hypothesized theories for the development of this cancer. The periferical production of estrogen in the fat could be a factor. Moreover, overweight usually is because of fatty tissue and not for hypertrophic musculature, so maybe these patients need proteins. Nevertheless, I agree with you that it could be a factor for explaining the results.

“ 5 – It is not described if the supplementation was made by regular diet or with some nutritional supplements. It is clear that biodisponibility of nutrients is different when supplemented isolated or in combination with other nutrients.”

To more clarify those aspects, we modified paragraph beginning line 89. Hope this clarification added help to better understand it.

All patients in the prehabilitation group received a nutritional education program involving food selection and meal planning patterns including an easy and feasible list of recipes for the homemade elaboration of protein supplements (mainly shakes and smoothies, always elaborated with natural and non-processed ingredients like fruits, vegetables, dried fruits, etc.) adapted to diabetic patients, if necessary. These recipes included around 20g of protein per day. All patients involved were previously instructed to take those oral protein supplements daily, always 30 min after exercise training, to enhance muscle hypertrophy. Those protein supplements prescribed didn’t substitute the normal protein intake during meals.

“6  - We believe that conclusions about the supplementation dose cannot be inferred because variables like biodisponibility and time of supplementation”

We add these 2 paragraphs addressing this commentary.

First, in line 226 we add:

We have also to remark that one of the reasons why we have not observed any benefit in our study, could be the period time during which the patients take the protein supplements. The mean time of protein supplementation in our series was 25 days, which could explain why no benefit was observed. However, in Kabata's et al study (17), with only 14 days, they observed beneficial effects in terms of morbidity and stability in nutritional parameters. The protein supplementation was also 20g of protein per day, although from a commercial preparation. We could think that the homemade natural shakes or smoothies are not as effective, perhaps due to bioavailability issues, as the commercial ones.

And in the weakness paragraph (line 246):

Finally, we must note also that we could not address aspects that should affect results like the biodisponibility of the nutrients.

Thank you very much for all your comments and hope that you will like the improved format of the article.

Reviewer 2 Report

+Most cases of endometrial cancer are overweight or obese. They donot suffer from protien deprivation . One could co=relate results with pre operative body weight and serum albunim levels. At times cases of endometrial cancer are operatedby open surgery,that can become another group for comparison. 

Author Response

Dear Reviewer,

Thank you very much for your valuable work and also for the comments to the article, that will help to improve the text.

Please, find here the answer to all your comments, one by one:

Comment 1:

“Most cases of endometrial cancer are overweight or obese. They donot suffer from protien deprivation.”

We agree with you, but in line with previous study our goal was to increase protein levels in women without protein deprivation. This objective is in line with the Kabata paper cited in the article, not going from bad to good, but from good to better. Maybe this cannot done if the protein levels are already good and could explain our results (although Kabata could demonstrate it).

Comment 2:

“One could correlate results with pre operative body weight and serum albumin levels.”

It is a good idea but it was not in our objectives. If you think it could add value to the paper we could study that, but we will need more time to send the results.

Comment 3:

“At times cases of endometrial cancer are operated by open surgery, that can become another group for comparison.”

We agree with you. We have data about patients operated by open surgery, but we decided not to include them for the following reasons:

  • Few patients operated by open surgery in the prehabilitation group, so impossible to arrive to any conclusions
  • The decrease in the albumin and protein serum levels is bigger in open surgery than in laparoscopy, independent of pathology (benign or malign), so it will add another variable that could confuse the results 

Thank you very much for all your positive and constructive comments, and hope you will like the improved format of the article.

Reviewer 3 Report

Paper could be readable, the aim is good and results are really interesting.

However, there are a lot of mistake that authors have to be correct before the possibility of publication. 

An editing of the written English had to be performed

Author write into the abstract that “this is a retrospective analysis” while in the M&M the paper was described as a prospective pilot observational study.

There is a mistake into le references list. They write “(Lancet)” instead of the reference number 1 (section: introduction; line: 3)

The acronym “MUST” should be written in its extended form the first time that it appears in the text. 

When authors use number into the table, they have to chose if use point or comma to express decimal numbers.Paper could be readable, the aim is good and results are really interesting.

Author Response

Dear Reviewer,

Thank you very much for your valuable work and for the comments to the article, that will help to improve the text.

Please, find here the answer to all your comments, one by one:

Comment 1:

However, there are a lot of mistake that authors have to be correct before the possibility of publication. An editing of the written English had to be performed.

Answer 1:

We have edit and correct the written English of the paper.

Comment 2:

Author write into the abstract that “this is a retrospective analysis” while in the M&M the paper was described as a prospective pilot observational study.

Thanks for the comment. You are right and we have changed the sentence to:

A prospective study including patients who underwent laparoscopy for endometrial cancer was conducted.

Comment 3:

There is a mistake into le references list. They write “(Lancet)” instead of the reference number 1 (section: introduction; line: 3)

Corrected. Now (Lancet) is reference number 2.

Comment 4:

The acronym “MUST” should be written in its extended form the first time that it appears in the text. 

Rephrased the sentence. Now we can read:

Patients in the prehabilitation program were screened for malnutrition with the Malnutrition Universal Screening Tool (MUST) test,

Comment 5:

When authors use number into the table, they have to chose if use point or comma to express decimal numbers. Paper could be readable, the aim is good and results are really interesting.

Thanks for your appreciation. We choosed to use points to express decimal numbers, but there were some mistakes with that in table 2. Now they are solved.

Thank you very much for all your suggestions and constructive reading!

Round 2

Reviewer 3 Report

The revised paper is acceptable in the present form.

I suggest author to perform a check of the style and language.